# High-Temperature-Annealed Multi-Walled Carbon Nanotubes as High-Performance Conductive Agents for LiNi$_{0.5}$Co$_{0.2}$Mn$_{0.3}$O$_2$ Lithium-Ion Batteries

Ziting Guo, Shengwen Zhong *, Mihong Cao, Zhengjun Zhong, Qingmei Xiao, Jinchao Huang and Jun Chen *

Key Laboratory of Power Battery and Materials of Jiangxi Province, School of Materials and Chemical Engineering, Jiangxi University of Science and Technology, Ganzhou 341000, China
* Correspondence: zhongshw@126.com (S.Z.); chenjun@jxust.edu.cn (J.C.)

**Abstract:** In this work, the high yield of MWNTs was prepared by chemical vapor deposition (CVD) method, followed by annealing at 2000–2800 °C, and the effects of high annealing temperature on metal impurities and defects in multi-walled carbon nanotubes (MWNTs) was explored. Furthermore, the annealed MWNTs were dispersed using a sand mill to make a conductive slurry, and finally the cathode LiNi$_{0.5}$Co$_{0.2}$Mn$_{0.3}$O$_2$ was added to the assembled batteries, and the application of MWNTs (slurry) as conductive agents in LiNi$_{0.5}$Co$_{0.2}$Mn$_{0.3}$O$_2$ (NCM) cathode materials by sand-mill dispersion on the performance of lithium-ion batteries was investigated. The results indicate that high temperature annealing can effectively remove the residual metal impurities from MWNTs and the defects in MWNTs gradually decreases as the temperature rises. In addition, 2 wt% of MWNTs (slurry) in LiNi$_{0.5}$Co$_{0.2}$Mn$_{0.3}$O$_2$ is sufficient to form an electronically conductive network; as a result, the electronic conductivity and the high rates performance of the LiNi$_{0.5}$Co$_{0.2}$Mn$_{0.3}$O$_2$ batteries were greatly improved. The LiNi$_{0.5}$Co$_{0.2}$Mn$_{0.3}$O$_2$ battery with MWNTs slurries annealed at 2200 °C as a conductive additive displays the highest initial discharge capacity of 173.16 mAh·g$^{-1}$ at 0.1 C. In addition, after 100 cycles, a capacity retention of 95.8% at 0.5 C and a discharge capacity of 121.75 mAh·g$^{-1}$ at 5 C were observed. The multi-walled carbon nanotubes used as conductive agents in LiNi$_{0.5}$Co$_{0.2}$Mn$_{0.3}$O$_2$ (NCM) cathode materials show excellent battery behaviors, which would provide a new insight for the development of high-performance novel conductive agents in lithium-ion batteries.

**Keywords:** multi-walled carbon nanotubes (MWNTs); annealing temperature; metal impurities; LiNi0.5Co0.2Mn0.3O2 cathode; lithium-ion battery

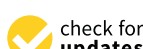



## 1. Introduction

Lithium-ion batteries have been widely used in consumer electronics because of their high operating voltage, high specific capacity, high output power, fast charging rate and long cycle life, as well as having no memory effect [1]. However, in recent years, with the booming development of electric vehicles, the shortcomings of power batteries have been gradually exposed, for example, low energy and insufficient power density [2]. In addition, the occurrence of electrochemical reactions in lithium-ion batteries requires electrons and lithium ions to reach the surface of the active material at the same time [3], so electrons can participate in the electrochemical reactions in time [4] in order to achieve good performance of the cathode active material [5,6]. Therefore, adding conductive agents with higher conductivity and reducing the amount of conductive agents is considered an effective strategy to improve the performance of lithium-ion batteries. However, carbon nanotubes can be used as an ideal carbon-based conductive additive for lithium-ion batteries, which is attributed to the formation of a three-dimensional high conductivity network with a low permeability threshold in the electrode with only a small amount of addition, ensuring the battery energy density while improving its electrochemical performance [7,8].

The discovery of carbon nanotubes and their unique physical properties have attracted widespread attention and interest from researchers. Carbon nanotubes have a high aspect ratio and a large specific surface area, and as one-dimensional nanomaterials, they are lightweight and have a perfectly connected hexagonal structure with many unusual mechanical, electrical and chemical properties. In recent years, with the deepening of carbon nanotubes and their nanomaterials research, their broad application prospects have been constantly shown. For example, due to their excellent conductive properties, MWNTs have been used as conductive agents in lithium-ion batteries and have the potential to replace conventional conductive agents, such as carbon black and Super P (SP) [9–11]. In addition, the electrical conductivity of MWNTs and the three-dimensional electron-conducting network both improve the stripping effect between the particles due to the volume change during charging and discharging [12], which affects the electrochemical performance of the cell.

Although CNTs have many advantages, they have two "fatal" disadvantages. First, the untreated carbon nanotubes will retain more metal impurities, so that their application in lithium ion batteries will increase internal friction, increase internal resistance in the battery and have great safety risks, as well as even being able to lead to the battery short circuit and spontaneous combustion. Secondly, the high aspect ratio of the one-dimensional structure and the strong van der Waals forces of the C-C bond make it very easy to agglomerate and difficult to disperse. If it is not well dispersed, it will cause higher internal resistance and higher internal consumption when used as a conductive agent in lithium-ion batteries, thus reducing the conductivity of the battery. To address these two issues, many studies have already been carried out. For example, Chung et al. [13] and Murdie et al. [14] investigated the effects of high temperature annealing and graphitization on the structure of carbon brazing dimensions. Lambert et al. [15] demonstrated that catalytic metal particles can be removed by annealing the synthesis of carbon nanotubes above the evaporation temperature of the metal catalyst. Andrews et al. [16] also reported high temperature annealing of carbon nanotubes between 1600–3000°C, and although it was not studied as a conductive agent on lithium-ion batteries, the results showed that high temperature annealing does improve the defects and structure of CNTs and is a low-cost and commercially feasible method for purifying and sorting MWNTs. In addition, Harald et al. [17] reported a process for dispersing carbon nanotubes using ultrasound, which can cause damage to the walls of the carbon nanotubes as the ultrasound and sonication time increases. Suyoung et al. [18] also reported the effect of mechanical grinding and grinding time on the dispersion of carbon nanotubes under different processes. Additionally, other research has also improved the dispersion of carbon nanotubes by sonication, coating and surfactants [19–22]. However, these modification techniques require advanced instruments and the yield of the resulting products is low. Currently, chemical vapor deposition (CVD) is the dominant method for the mass production of carbon nanotubes [23], but the carbon nanotubes made by this method have a high residual metal catalyst and defects. Typically, impurities in MWNTs are removed using chemical methods; a simple process that is widely distributed in industry, but which relies on the higher chemical stability of carbon nanotubes, relative to the metallic impurities and amorphous carbon in them [24]. In addition, chemical method usually pollutes the environment and requires the stability of carbon nanotubes. Unlike chemical methods, physical purification causes low damage to carbon nanotubes and is environmentally friendly [25]. By high temperature annealing the carbon nanotubes, the nanoscale effect can be used to gasify the metal particles in the carbon nanotube to remove impurities while maintaining the structure of the carbon nanotube and even reducing the defect [26]. In addition, in order to improve the electrical conductivity of carbon nanotubes, many researchers have attempted doping introduced defects to improve the diffusion rate of lithium ions [27], and Shimoda et al. [28] reported an increase in capacity attributed to lithium-ion diffusion into the interior of SWNTs through inception and sidewall defects. This indicates that the defect facilitates the migration and diffusion of lithium ions.

Through the batch preparation of MWNTs by CVD method, the high-yield MWNTs are easier to disperse them using a sand mill, and it is found that the high-temperature annealing method can not only effectively remove metal impurities but also reduce the defects of MWNTs. Therefore, the MWNTs conductive slurry formed through the above two steps should be the conductive material that can enhance the cathode of the lithium-ion battery. Although high temperature annealing of MWNTs has been widely studied, the slurry of MWNTs prepared after high temperature annealing of MWNTs is rarely studied as a conductive additive for the cathode, especially in batteries with nickel-rich ternary cathode materials. Here, in this work, a high yield of MWNTs was prepared by chemical vapor deposition (CVD) method, followed by annealing at 2000–2800 °C, and the effects of a high annealing temperature on metal impurities and defects in multi-walled carbon nanotubes (MWNTs) was explored. The results show that the content of metallic impurities in MWNTs gradually decreases as the annealing temperature increases, and the degree of graphitization increases and the defects decrease. Among them, the slurry made from MWNTs after high temperature annealing at 2200 °C applied to $LiNi_{0.5}Co_{0.2}Mn_{0.3}O_2$ cathodes showed the best battery performance, such as higher capacity, better rate capability and cycle behaviors, which would provide a new insight for the development of high-performance novel conductive agent in lithium-ion batteries.

## 2. Experimental

### 2.1. Preparation of Multi-Walled Carbon Nanotubes

Iron nitrate, aluminum nitrate and citric acid monohydrate were weighed as the solvents by mass of 20.2, 18.76, and 21.01 g, respectively. The weighed materials were added into 30 mL of deionized water and stirred until completely dissolved. Consequently, the mentioned solvents were mixed and stirred for 12h. After the sol–gel preparation, the obtained crude product was dried in a 120 °C drying tank for 12 h. After that, the dried powder was first ground, heated to 200 °C for 120 min at 5 °C/min in a tube furnace, then heated to 800 °C for 60 min at 5 °C/min. The catalyst was then cooled to room temperature and ground in an agate mortar and finally sieved through a 300 mesh sieve and placed in a tube furnace for the preparation of MWNTs by CVD method: Argon was used as the protective gas, hydrogen as the reducing gas and propane as the carbon source. The furnace was first fed with argon at 500 sccm and then heated to 680 °C at 5 °C/min for 140 min. After heating the furnace for 30 min, the hydrogen was passed through at 200 scmm for 100 min, then the hydrogen flow rate was increased to 500 sccm for 20 min, and then to 100 sccm and 300 sccm, respectively, while opening propane was at 300 sccm for 120 min and finally argon was restored to 500 sccm until it reached room temperature.

### 2.2. High-Temperature-Annealed Multi-Walled Carbon Nanotubes

The synthesized carbon nanotubes were placed in a graphitization furnace for high temperature annealing at 2000–2800 °C. $Ar_2$ was passed as a protective gas during the annealing process, and the temperature was ramped up to 1200 °C at 40 °C/min, then 20 °C/min to different annealing temperatures and held for 180 min; the obtained samples were labelled as CNT-2000 °C, CNT-2200 °C, CNT-2400 °C, CNT-2600 °C and CNT-2800 °C, and the un-annealed MWNTs was labelled as CNT.

### 2.3. Preparation of Multi-Walled Carbon Nanotube Conductive Slurry

N-methyl-pyrrolidone (NMP), MWNTs and dispersant PVP were mixed at a solid content of 95:4:1 for 20 min, during which 1600 g of zirconium beads with size of 0.8–1 μm were added in a sand mill (Horizontal sand mill, NT-0.6L, LONGLY). The mixed MWNTs crude slurry was poured into a sand mill and sanded at 1600 r/min for 30–40 min.

### 2.4. Assemble of Batteries Using Multi-Walled Carbon Nanotube Slurry as Conductive Agent

The $LiNi_{0.5}Co_{0.2}Mn_{0.3}O_2$ cathode material, binder PVDF and NMP were mixed at a mass ratio of 96:6:2 to obtain a solid content of 50% mixture, which was put into the

homogenizer and continuously stirred for 30 s at 800 r/min, and then the speed was automatically increased to 2000 r/min for 20 min to obtain the evenly dispersed electrode slurry. Then, the slurry was evenly coated onto the aluminum collector using a coating machine, and the coating thickness of the slurry was 0.019 mm. Then, the obtained electrode sheet was dried at 120 °C for 24 h. The CR2032 button cell was assembled using lithium metal as the anode in a glove box filled with argon gas under a moisture and oxygen content of less than 0.01 ppm. The batteries with $LiNi_{0.5}Co_{0.2}Mn_{0.3}O_2$ cathodes using annealing of MWNTs with different temperatures were labelled as NCM-CNT(2000 °C), NCM-CNT(2200 °C), NCM-CNT(2400 °C), NCM-CNT(2600 °C) and NCM-CNT(2800 °C), respectively. Cells in which SP is used as a conductive agent instead of MWNTs slurry were labelled as NCM-SP, and the MWNTs (slurry) without impurity removal were labeled as NCM-CNT, ensuring that all other conditions remained the same.

*2.5. Characterization and Performance Testing*

The morphology of the carbon nanotubes was characterized by scanning electron microscopy (SEM, EVO/MA10, ZEISS, Oberkochen, Germany). Defects in MWNTs were characterized by Raman spectroscopy (Horiba HR800, Kyoto, Japan) with a laser emission wavelength of 514 nm. The structure, degree of graphitization and metallic impurities of the MWNTs were characterized by X-ray diffraction (XRD, Rigaku MiniFlex 600, Tokyo, Japan) with a scanning range of 10° to 80° and a speed of 10°/min. The metal impurity content of the MWNTs after annealing at different temperatures was characterized by inductively coupled plasma emission spectrometry (ICP, Agilent 5900, Palo Alto, CA, USA) with an optical resolution of <0.007 nm and a CCD detector covering the wavelength range of 167–785 nm. The distribution of MWNTs in the active material was observed by SEM. The batteries were tested for first charge/discharge, multiplier and cycling performance using the NEW-ARE battery test apparatus at voltages ranging from 2.7–4.2 V. AC impedance (EIS) was carried out on an electrochemical workstation (PGSTAT101, Metrohm Autolab, Utrecht, Switzerland) with a frequency range of $10^{5-}$ 0.1 Hz and an amplitude of 5 mV. All test characterizations were carried out at room temperature.

**3. Result and Discussion**

*3.1. Inductively Coupled Plasma Emission Spectroscopy (ICP) Analysis*

The ICP of MWNTs annealed at different temperatures for 180 min is shown in Figure 1. As the annealing temperature increases, the metal impurities in MWNTs gradually decrease. It is worth noting that the content of iron impurities drops sharply at 2200 °C, with a smaller drop in the content of iron and aluminum impurities after 2200 °C. The inset in Figure 1 shows the amount of metal catalyst remaining in the MWNTs at different annealing temperatures after empty firing at 800 °C, where the metal catalyst is no longer visible in the porcelain boat at CNT-2200 °C, while some catalyst remains at CNT-2000 °C, which is consistent with the ICP data. As can be seen more specifically in Table 1, the content of metal impurities drops below 200 ppm at temperatures up to 2400 °C and below 100 ppm for both Fe and Al when temperatures reach 2800 °C. This suggests that high temperature annealing can achieve the effective removal of metal impurities from MWNTs.

**Table 1.** Impurity content of MWNTs annealed at different temperatures for 180 min by ICP.

| Sample Number (ppm) | Fe | Al |
|:---:|:---:|:---:|
| CNT | 81700 | 29800 |
| CNT-2000 °C | 56400 | 1400 |
| CNT-2200 °C | 300 | 189 |
| CNT-2400 °C | 150 | 132 |
| CNT-2600 °C | 118 | 125 |
| CNT-2800 °C | <100 | <100 |

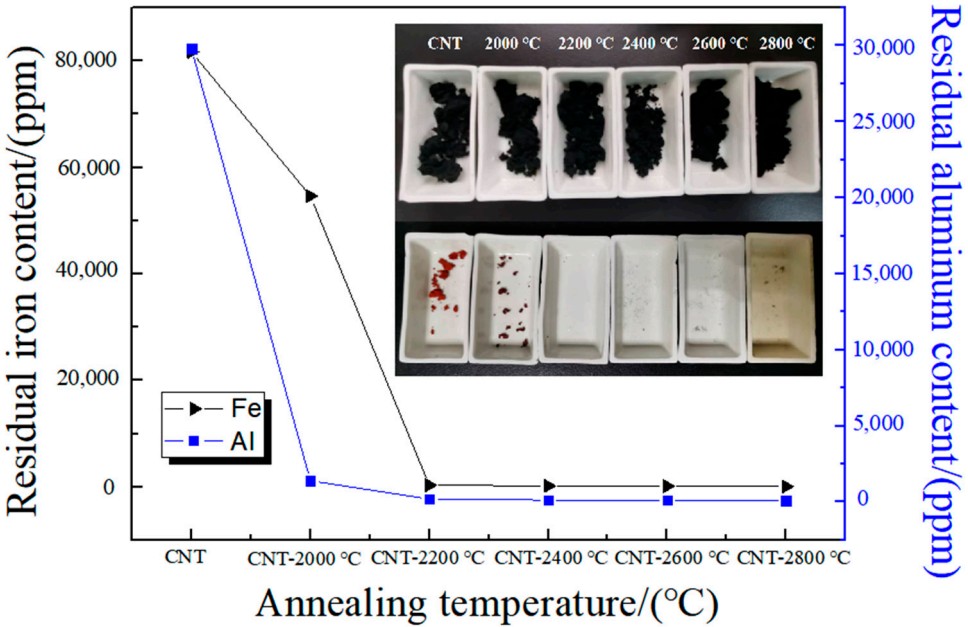

**Figure 1.** ICP of MWNTs annealed at different temperature for 180 min, the inset shows MWNTs with different annealing temperatures before and after air firing at 800 °C.

### 3.2. X-ray Diffraction (XRD) Analysis

The XRD of MWNTs annealed for 180 min at different temperatures is shown in Figure 2a. The 002 peak does not change significantly for the CNT-2000 °C and CNT-2200 °C samples, while for the CNT-2400 °C sample, the 002 peak gradually increases, and as the temperature continues to rise, the 002 peak gradually increases and becomes sharper. The 002 peak is highest for the CNT-2800 °C sample, and the characteristic peaks of carbon nanotubes 100 and 004 are also more pronounced at 2800 °C [29]. The result indicates that the MWNTs are more graphitized and highly ordered at higher temperatures. It is clear from Figure 2b that the CNT contains other impurity peaks, which were calibrated by PDF card, and it can be confirmed that the prepared MWNTs contain metal impurities, composing mostly iron carbide and alumina, which have been calibrated in the figure. In addition, a splitting of the 100 peak in the CNT was found, which was caused by the overlap of the iron carbide phase with the carbon nanotube peak. Combined with Figure 2a, it can be seen that when the annealing temperature is at 2000 °C, these metallic impurity phases almost disappeared for the CNT-2000 °C sample, while as the temperature increases, the 100 peak gradually increases. This is consistent with previous conclusions reached through ICP.

### 3.3. Raman Spectroscopy

High temperature annealing can not only remove metal impurities but can also reduce defects in MWNTs. Raman spectra for the four samples of CNT, CNT-2000 °C, CNT-2400 °C and CNT-2800 °C were investigated, as shown in Figure 3a. The strong peak near 1580 cm$^{-1}$ corresponds to the G-peak, which is used to reflect the degree of graphitization of the carbon nanotubes, and the strong peak near 1350 cm$^{-1}$ corresponds to the D-peak which is used to reflect the disorderly fine graphite structure, i.e., the disorganization of the carbon nanotube structure. It is found that as the annealing temperature increases the D-peak gradually decreases, indicating an increase in the structural order of the MWNTs, while the G-peak gradually increases, indicating a better graphitization of the MWNTs and fewer defects. The ID/IG ratios calculated after extracting the areas of the D and G peaks are plotted in Figure 3b. The ID/IG values were 0.68, 0.556, 0.499 and 0.473 for CNT, CNT-2000 °C, CNT-2400 °C and CNT-2800 °C, respectively. It can be found that the ID/IG ratio decreases as the temperature increases, which means that the higher the graphitization

of MWNTs, the lesser the defects, and this conclusion again confirms the previous XRD. Moreover, it is fully stated that high temperature annealing can not only remove metal impurities in MWNTs but also reduce their defects.

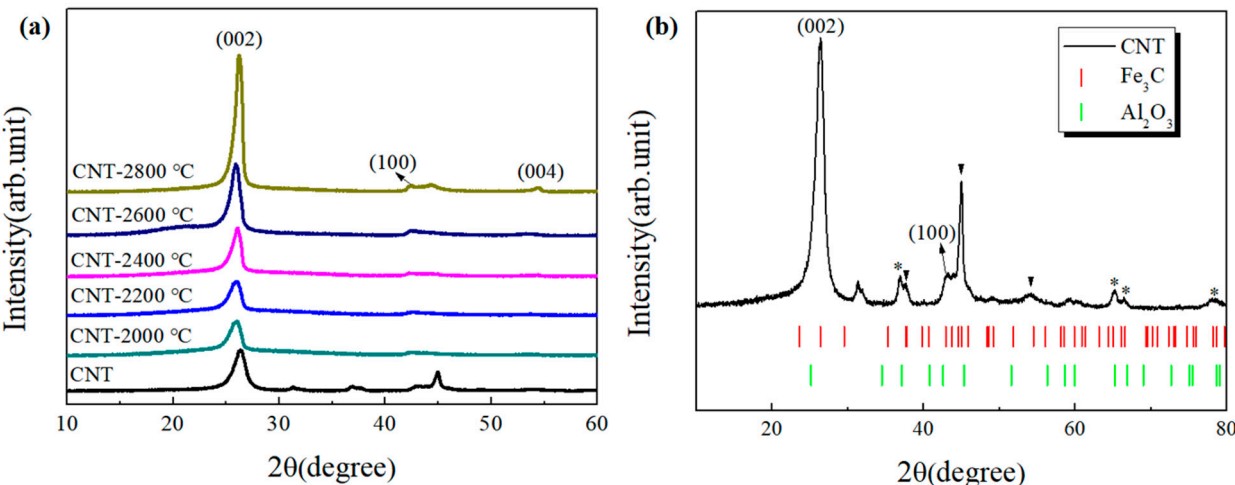

**Figure 2.** (**a**) XRD of MWNTs annealed at different temperature for 180 min, (**b**) Impurity distribution and peak level calibration of undiluted MWNTs. Peaks with ▼ mark are iron carbide, while peaks with ∗ mark are aluminum oxide.

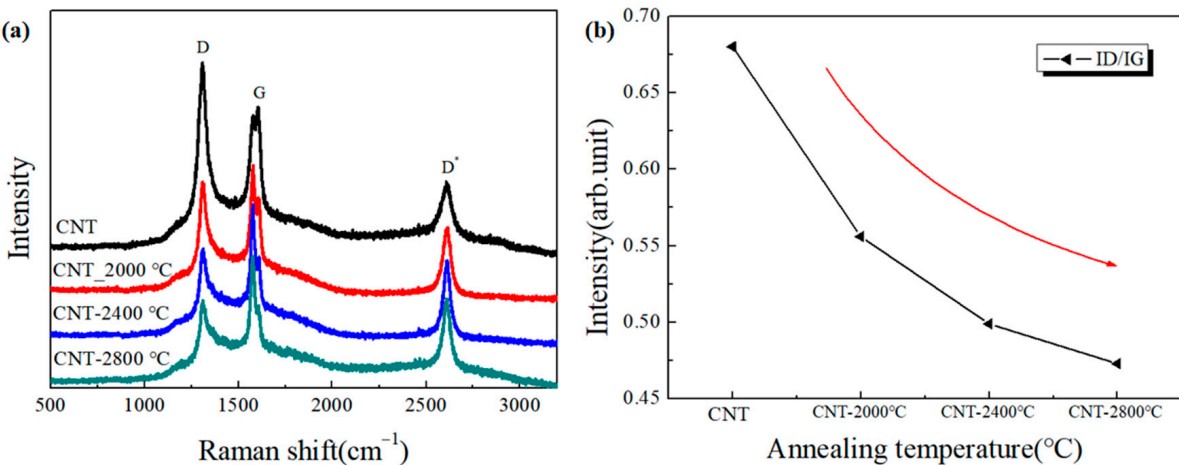

**Figure 3.** (**a**) Raman curves of MWNTs annealed at different temperatures for 180 min, (**b**) Ratio curves of D and G peaks in Raman.

### 3.4. Scanning Electron Microscopy (SEM) Analysis

The SEM images of the CNT, CNT-2000 °C, CNT-2400 °C and CNT-2800 °C four samples were observed, as shown in Figure 4a–d. Figure 4a shows that the CNT contains more metal impurities (shown by arrows), and the morphological features of MWNTs entangled with each other can be observed. As can be seen from Figure 4b,c, the samples CNT-2000 °C and CNT-2400 °C showed a significant decrease in metal impurities with the increase in annealing temperature (shown by arrows), but the agglomeration phenomenon was still serious. When the annealing temperature was increased to 2800 °C, the metal impurities in MWNTs basically disappeared for the CNT-2800 °C sample, and the smooth and flat walls of MWNTs could be seen, as shown in Figure 4d. The result indicated that the agglomeration of MWNTs before and after annealing was very severe. On the contrary, the sanded MWNTs are not entangled with each other and the agglomeration phenomenon is greatly improved, as shown in Figure 4e. In addition, as shown in Figure 4f, afterwards,

the annealed and dispersed MWNTs slurry was added into the cathode material to build a continuous conductive network with point-line contact between the MWNTs and the cathode material, which increased the contact area to improve the lithium ion migration rate and interconnected the individual cathode particles. The results of the SEM images corroborate mutually with the previous results of ICP.

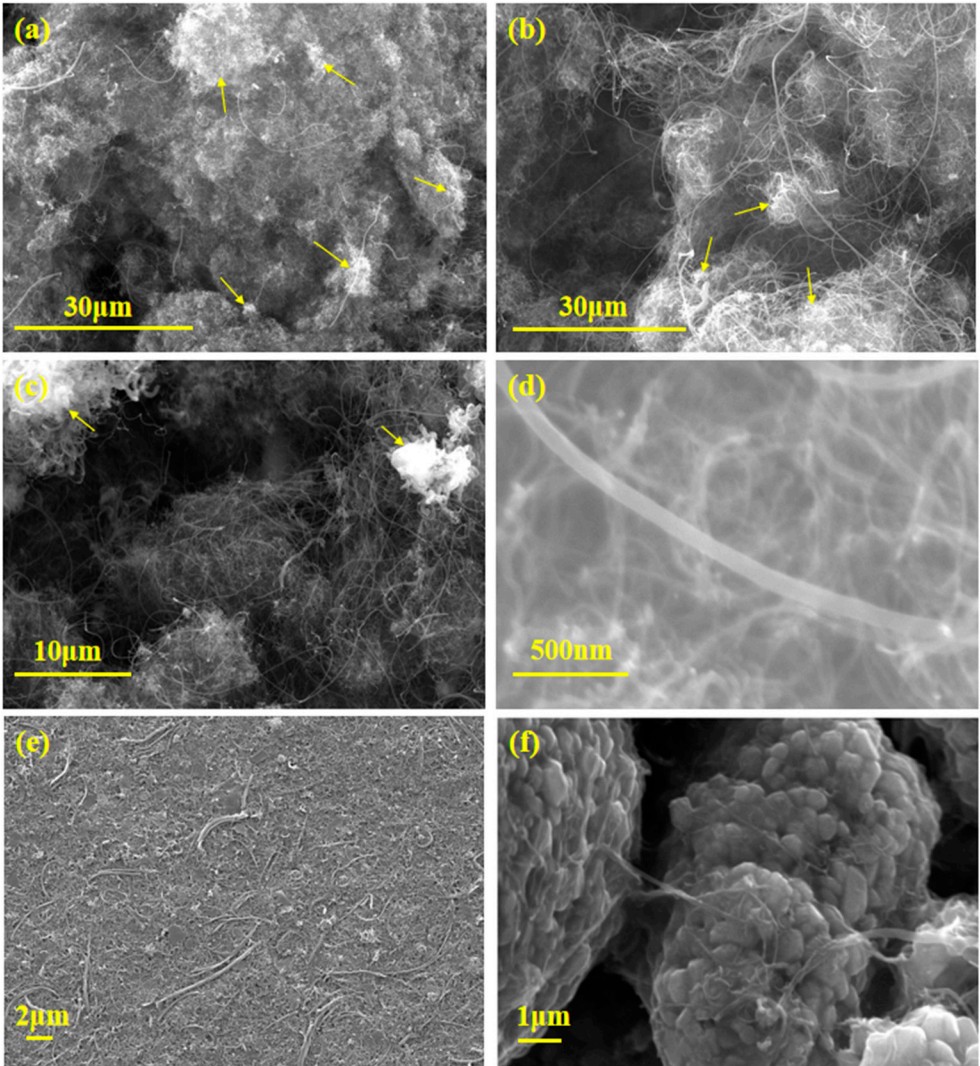

**Figure 4.** Characterization of MWNTs, SEM images of (**a**) undiluted MWNTs; (**b**) MWNTs annealed at 2000 °C; (**c**) MWNTs annealed at 2400 °C; (**d**) MWNTs annealed at 2800 °C; (**e**) MWNTs after sanding and dispersion; (**f**) annealed MWNTs slurry cathode sheet.

### 3.5. Research in Transmission Electron Microscopy

The TEM images of the CNT, CNT-2200 °C, CNT-2400 °C and CNT-2800 °C are shown in Figure 5a–d. The metal catalyst in the CNT sample is bounded with the MWNTs tube that is shown by arrows in Figure 5a. The walls of the MWNTs were bent and aligned irregularly, indicating great defects in the MWNTs at this point, which is shown by arrows in Figure 5b. However, the metal catalyst inside the MWNTs tube disappears while increasing the annealing temperature (shown in Figure 5c,e,g). The carbon atoms and the graphene layer on the walls of MWNTs were gradually ordered and became regular while increasing the annealing temperature, which revealed a great decrease in the defects of MWNTs (shown in Figure 5d,f,h). Notably, the planar spacing between both graphene layers within the MWNTs tubes was measured at 0.34–0.35 nm for the samples, which corresponds to the

planar spacing value of multi-walled carbon nanotubes [30]. The TEM images mutually corroborate the ICP results presented previously in Figure 1.

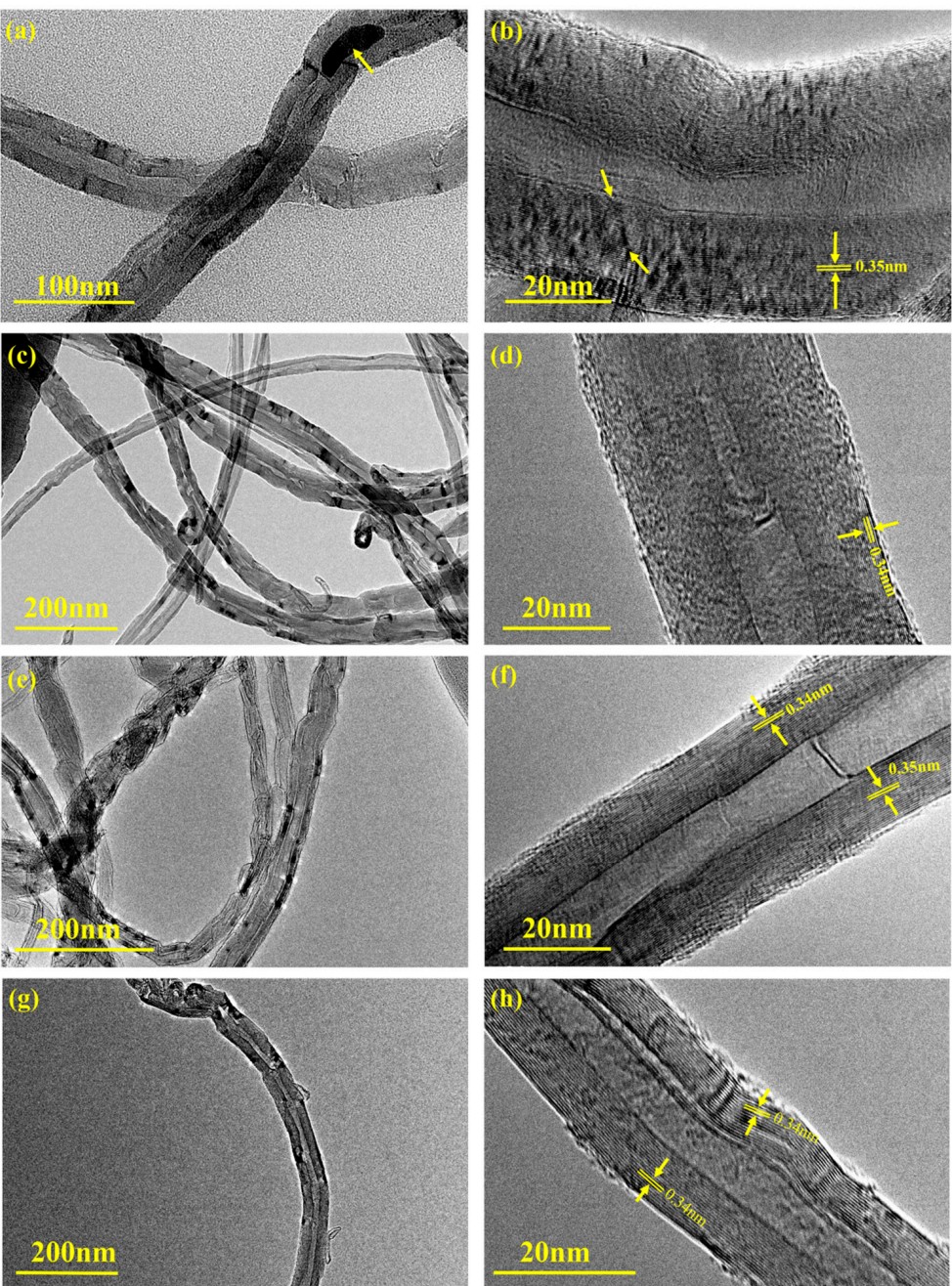

**Figure 5.** Characterization of MWNTs, TEM images of (**a**,**b**) undiluted MWNTs; (**c**,**d**) MWNTs annealed at 2200 °C; (**e**,**f**) MWNTs annealed at 2400 °C; (**g**,**h**) MWNTs annealed at 2800 °C.

### 3.6. Electrochemical Analysis

Figure 6a shows the initial charge/discharge curves of different batteries at 0.1 C (1 C = 155 mAh g$^{-1}$) between 2.75–4.2 V. It can be said that the initial discharge capacities were 164.29, 173.16, 167.75, 171.57, 172.21, 162.25 and 102.78 mAh·g$^{-1}$, for the NCM-CNT (2000 °C), NCM-CNT (2200 °C), NCM-CNT (2400 °C), NCM-CNT (2600 °C), NCM-CNT (2800 °C), NCM-SP and original carbon nanotubes without impurity removal (NCM-CNT) batteries, respectively. Among them, the NCM-CNT (2200 °C) battery display the highest initial discharge capacity. In contrast, the NCM-SP battery using SP as the conductive

agent had the lowest initial discharge capacity, which was attributed to its low diffusion rate of lithium ions due to the small contact area between SP and the active material. It is worth noting that all the CNT-2000 °C, CNT-2200 °C, CNT-2400 °C, CNT-2600 °C and CNT-2800 °C samples have an optimizing effect on the conductivity of $LiNi_{0.5}Co_{0.2}Mn_{0.3}O_2$ (NCM), which is attributed to the elimination of metal impurities by high temperature annealing and the better dispersion obtained by sand grinding treatment, which enables MWNTs to connect more active particles, thus building a continuous conductive network, increasing the contact area with the active particles and ultimately improving the migration rate of lithium ion.

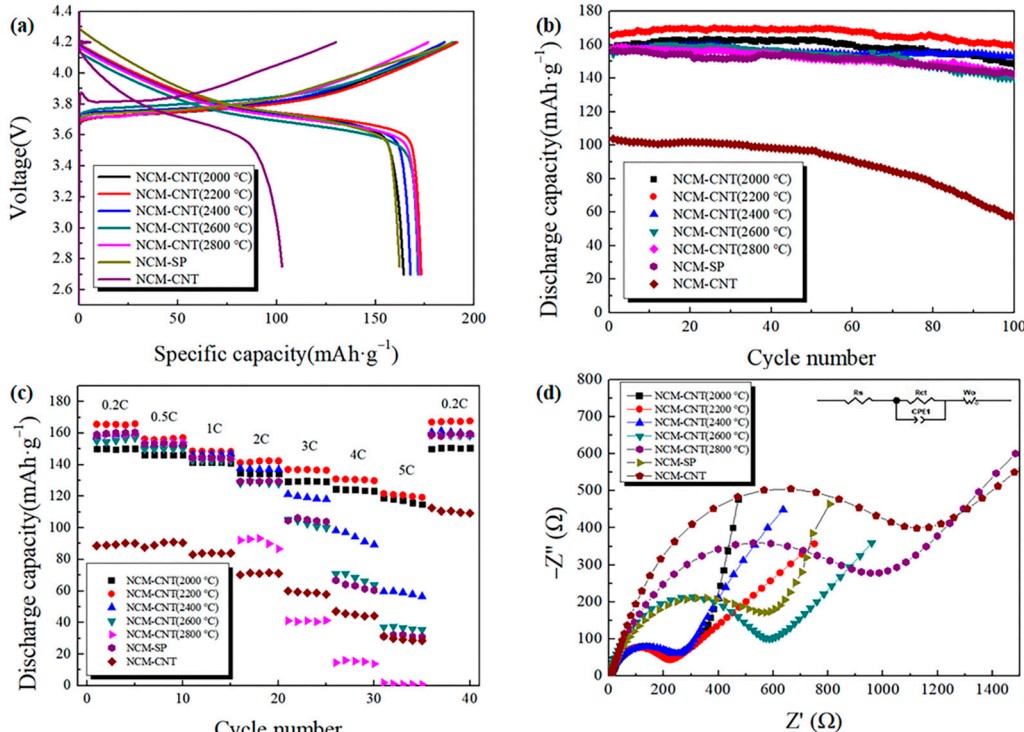

**Figure 6.** Electrochemical properties of $LiNi_{0.5}Co_{0.2}Mn_{0.3}O_2$ cathodes using MWNTs slurry made by abrasive dispersion at different annealing temperatures. (**a**) Initial galvanostatic charge–discharge profiles at 0.1 C; (**b**) cycling performance at 0.5 C; (**c**) rate performance of all samples; (**d**) the EIS data of all samples.

The cycling behaviors of the seven samples charged and discharged at 0.5 C for 100 times is investigated, as shown in Figure 6b. Among them, the NCM-CNT (2200 °C) battery presents the best cycle performance (red curve), and the initial discharge capacity is 165.58 mAh·g$^{-1}$ at 0.5 C with a capacity retention rate of 95.8% after 100 cycles. In addition, the initial capacities of NCM-CNT (2000 °C), NCM-CNT (2400 °C), NCM-CNT (2600 °C), NCM-CNT (2800 °C), NCM-SP and NCM-CNT batteries were 158.71, 156.39, 154.41, 154.74, 156.26 and 102.69 mAh·g$^{-1}$ with capacity retention rates of 93.6%, 97.8%, 90.7%, 90.1%, 91.28% and 54.92%, respectively. The results show that the cycle performance of the batteries participating in carbon nanotubes as a conductive agent is better than that of SP as a conductive agent. In addition, the performance of the NCM-CNT cycle is the worst because there are a lot of impurities, such as metal catalysts in the original MWNTs, the polarization of the battery is very serious during the charge–discharge process, so the cycle efficiency is low.

Figure 6c shows the discharge capacity of the batteries at different current rates (0.2 C, 0.5 C, 1 C, 2 C, 3 C, 4 C, 5 C, 0.2 C). It is found that the capacities of NCM-CNT (2400 °C), NCM-CNT (2600 °C), NCM-CNT (2800 °C), NCM-SP and NCM-CNT batteries decreases sharply with increasing current rate and even the NCM-CNT (2800 °C) battery has almost

no reversible capacity at a charge rate of 5 C, and the NCM-CNT battery has the lowest capacity under several current rates, indicating that MWNTs without impurity removal are not conducive to improving the conductivity of the cathode material. In contrast, the NCM-CNT (2200 °C) battery showed excellent rate performance; all the discharge capacities at different rates were higher than those of other NCM-CNT batteries. At the largest rate of 5 C, a high capacity of 121.75 mAh·g$^{-1}$ was observed for the NCM-CNT (2200 °C) battery, which was 90.64 and 90.28 mAh·g$^{-1}$ higher than that of NCM-SP and NCM-CNT battery and was superior to the other batteries at different current rates. It is noteworthy that although the samples with higher annealing temperatures (CNT-2400 °C, CNT-2600 °C, CNT-2800 °C) had fewer defects and higher graphitization of MWNTs, they all exhibited lower electrochemical performance when applied as conductive agents to LiNi$_{0.5}$Co$_{0.2}$Mn$_{0.3}$O$_2$ than the NCM-CNT (2200 °C) samples. This may be due to the fact that a portion of the MWNTs with very few defects acted as the anode, and some of the lithium ions were embedded in the MWNTs during the charging and discharging process, resulting in a lower capacity [31].

Electrochemical impedance spectroscopy (EIS) spectra of different batteries were investigated to understand the effect of different high-temperature-annealed MWNTs as a conducting agent on the performance of LiNi$_{0.5}$Co$_{0.2}$Mn$_{0.3}$O$_2$ cathode compared to SP. As shown in Figure 6d and Table 2, the inset is the equivalent circuit used for the simulation analysis of the EIS data, the Nyquist curves for all samples consist of a semicircle in the high- or mid-frequency region and a straight line in the low-frequency region. The intercept of the curve with the horizontal axis Z′ represents the ohmic resistance R1, which mainly represents the electrolyte, diaphragm and active substance of the cell; the arc in the mid-frequency region corresponds to the charge transfer impedance R2 on the surface of the active substance; the diagonal line in the low-frequency region corresponds to the Warburg impedance, which is the impedance caused by the diffusion of lithium ions in the active substance [32]. It can be seen that although the NCM-CNT (2000 °C), NCM-CNT (2200°C), NCM-CNT (2400 °C), NCM-CNT (2600 °C), NCM-CNT (2800 °C), NCM-SP and NCM-CNT batteries employ the same cathode electrolyte, they display different the SEI film impedance and charge transfer impedance. Among them, the NCM-CNT (2200 °C) battery presents the lowest SEI film impedance and charge transfer impedance, indicating that the lithium ion migration rate is highest at this point and the battery has lower internal consumption and better conductivity. However, the SEI film impedance and charge transfer impedance of the NCM-CNT (2000 °C) battery is similar to that of NCM-CNT (2400 °C) battery, but both of them are larger than that of NCM-CNT (2200 °C) battery. In addition, the sum of SEI film impedance and charge transfer impedance was greatest for the NCM-CNT batteries. The results corresponds well to cycle and rate behaviors, which indicates that the NCM-CNT (2200 °C) sample had the best electrical conductivity.

**Table 2.** Cell EIS parameters for MWNTs (slurry) cathodes with different annealing temperatures.

| Sample | R$_1$ (mΩ) | R$_2$ (mΩ) |
|---|---|---|
| NCM-CNT (2000 °C) | 6.972 | 190.6 |
| NCM-CNT (2200 °C) | 1.142 | 107.1 |
| NCM-CNT (2400 °C) | 0.261 | 129.6 |
| NCM-CNT (2600 °C) | 2.007 | 364.3 |
| NCM-CNT (2800 °C) | 6.571 | 298.9 |
| NCM-SP | 4.368 | 271.7 |
| NCM-CNT | 3.552 | 707 |

In order to investigate the effect of MWNTs (slurry) conductive agent annealed at different temperatures on the performance of LiNi$_{0.5}$Co$_{0.2}$Mn$_{0.3}$O$_2$ cathodes, the initial discharge curves of different batteries at different current rates (0.2 C, 0.5 C, 1 C, 2 C, 3 C, 4 C, 5 C) were analyzed, as shown in Figure 7a–f. It can be seen that the discharge voltage plateau and discharge capacity gradually decrease as the discharge rate increases. This

is due to the higher current rate which polarizes the cell and prevents the migration of lithium ions, thus reducing the conductivity of the cathode. Comparing the changes in the discharge capacity of the six batteries, the NCM-CNT (2200 °C) battery had the highest initial discharge capacity at all current rates, and its discharge plateau was stable with no distortion in the discharge curve. In contrast, the discharge curve of the NCM-CNT (2000 °C) battery is stable but its discharge capacity at different rates is lower than that of NCM-CNT (2200 °C) battery, probably because there are still more metal impurities in MWNTs, leading to higher internal consumption of the battery, while the discharge curves of NCM-CNT (2400 °C), NCM-CNT (2600 °C), NCM-CNT (2800 °C) and NCM-SP batteries showed distorted and unstable discharge curves at high current rates, and the discharge capacity also decreased substantially.

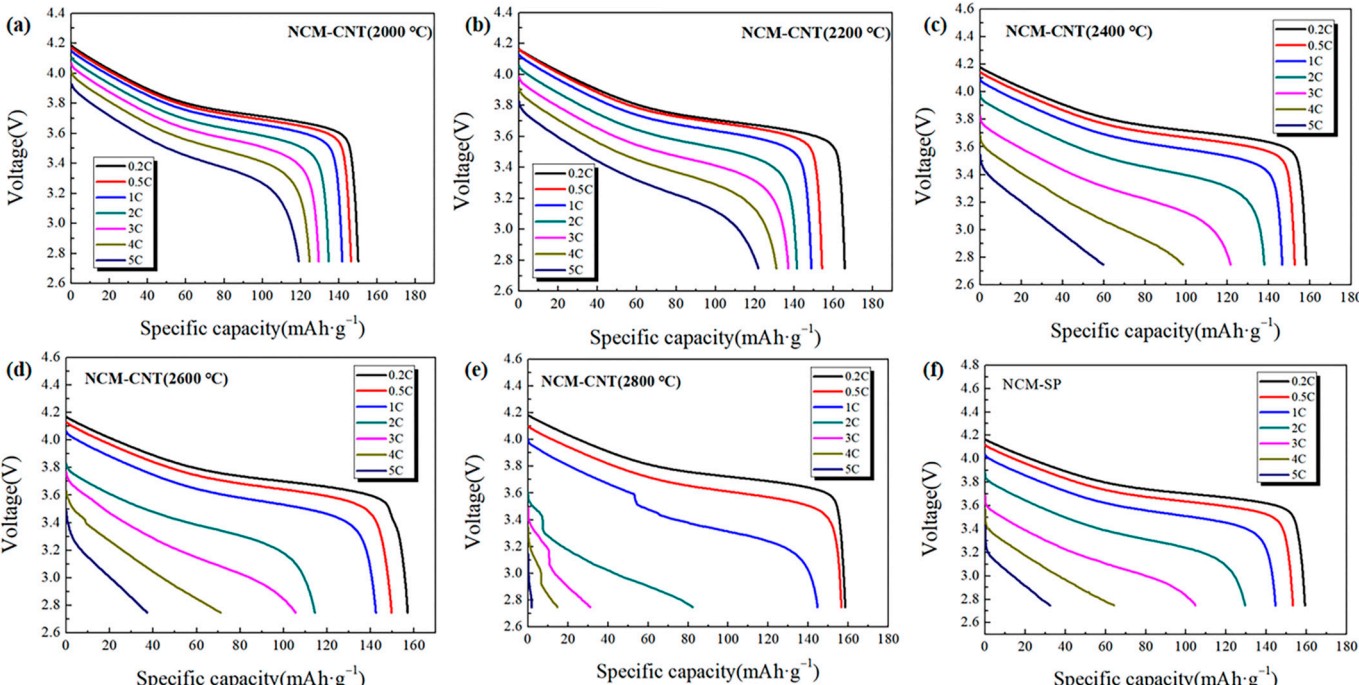

**Figure 7.** Initial discharge curves NCM-CNT (2000–2800 °C) and NCM-SP batteries at different rates of all samples. (**a**) is the first discharge curve of CNM-CNT(2000 °C) at different current rates; (**b**) is the first discharge curve of CNM-CNT(2200 °C) at different current rates; (**c**) is the first discharge curve of CNM-CNT(2400 °C) at different current rates; (**d**) is the first discharge curve of CNM-CNT(2600 °C) at different current rates; (**e**) is the first discharge curve of CNM-CNT(2800 °C) at different current rates; (**f**) is the first discharge curve of CNM-SP at different current rates.

After using MWNTs as a conductive agent, the performance of the battery is significantly improved. This is mainly attributed to the mechanism of improving the conductivity caused by the different microstructures of MWNTs. First of all, the mechanism of carbon nanotubes as new carbon-based conductive agents whose electrical conductivity is superior to that of traditional conductive agents, such as carbon black and SP, is explained as follows. The unique one-dimensional structure of carbon nanotubes increases the contact area with the active material from "point-to-point" to "point-to-wire" contact, increasing the migration rate of lithium ions, reducing the internal resistance and the internal consumption of the battery, as shown in Figure 8a.

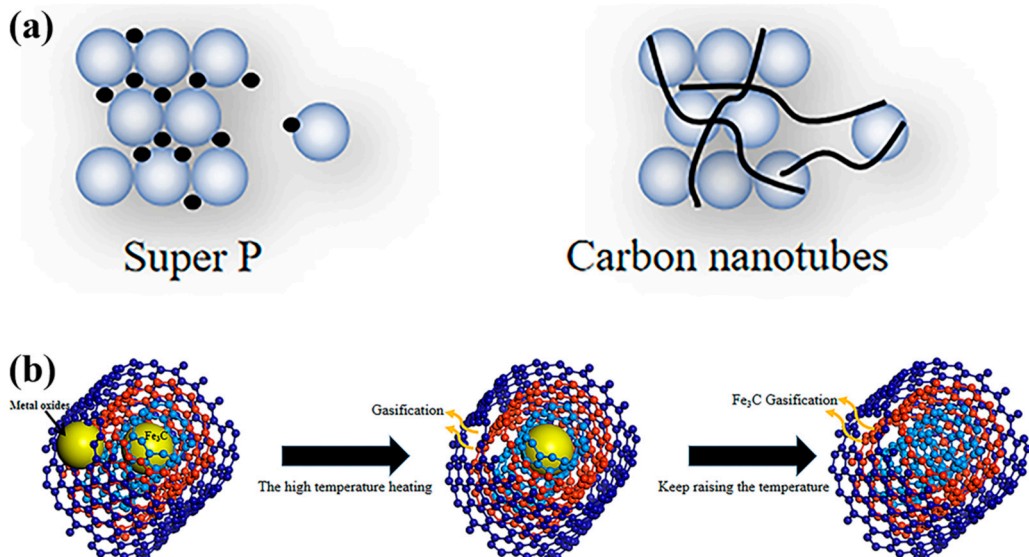

**Figure 8.** (**a**) Schematic diagram of Super P, CNTs and positive particles connection; (**b**) Schematic diagram of the mechanism of high-temperature purification of carbon nanotubes.

In addition, the mechanism of metal impurity removal by MWNTs is explained as follows. As the annealing temperature increases, the metallic impurities embedded in the pipe wall and inside the pipe will vaporize below their boiling points due to the nano-effect, and thus be eliminated along the MWNTs pipe or by opening the pipe wall, as shown in Figure 8b. As a result, a more perfect conductive network is built, ultimately improving the overall performance of the battery.

## 4. Conclusions

In this work, the method of high temperature annealing and sand grinding dispersion was adopted to solve the two main problems of MWNTs used in the battery for metal impurities and agglomeration, as well as to determine whether the metal impurities contained in the prepared MWNTs are mostly iron carbide and alumina. In addition, it is found that the high temperature annealing can not only remove metal impurities but also reduce the defects of MWNTs, and the sand grinding treatment can effectively disperse MWNTs. The dispersed MWNTs slurry applied to the $LiNi_{0.5}Co_{0.2}Mn_{0.3}O_2$ cathode can build a continuous and stable conductive network, thus improving the electrical conductivity of the cathode compared to the SP conductive agent. Notably, this work provides a simple and efficient and environmentally friendly method for preparing MWNTs (slurry) conductive agents and systematically studies the effect of slurry made of MWNTs at different annealing temperatures as conductive additives on the conductivity of the $LiNi_{0.5}Co_{0.2}Mn_{0.3}O_2$ cathode. Among them, the NCM-CNT (2200 °C) battery shows excellent electrochemical properties, which can be attributed to its rich defective pipe wall structure, assisting the transmission and diffusion of lithium ions. This work provides new ideas for the development of new high-performance CNTs-based conductive agents for lithium-ion batteries.

**Author Contributions:** Conceptualization, Z.G. and S.Z.; Methodology, Z.G. and M.C.; Software, S.Z.; Validation, Z.G. and Q.X.; Formal analysis, Z.Z.; Resources, Z.G.; Writing—original draft, Z.G.; Writing—review & editing, J.C.; Project administration, J.C.; Funding acquisition, J.H. All authors have read and agreed to the published version of the manuscript.

**Funding:** This work was supported by the National Natural Science Foundation of China (51874151), the Scientific Research Foundation for Universities from the Education Bureau of Jiangxi Province (GJJ170510), the Natural Science Foundation of Jiangxi Province (20151BBE50106) and the Jiangxi University of Science and Technology (NSFJ2014-G13, Jxxjbs12005).

**Institutional Review Board Statement:** Not applicable.

**Informed Consent Statement:** Informed consent was obtained from all subjects involved in the study.

**Data Availability Statement:** All the data and proofs have been provided in the article.

**Conflicts of Interest:** The authors declare no conflict of interest.

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
