# Peer review of "High-Temperature-Annealed Multi-Walled Carbon Nanotubes as High-Performance Conductive Agents for LiNi0.5Co0.2Mn0.3O2 Lithium-Ion Batteries"

_metals, doi:10.3390/met13010036_

Round 1
Reviewer 1 Report
In the manuscript titled “High-temperature annealed multi-walled carbon nanotubes as high-performance conductive agents for lithium-ion batteries” the author has done a good job explaining the manufacturing procedure and the results clearly with proper justification. The reviewer has a few questions,
1. The author claims the CVD procedure used produces high-yield MWNTs. Can the author quantify the claim?
2. Line 121 – “A certain mass of iron nitrate…”. Can the author specify the exact mass of each material used?
3. Line 152 – what was the coating thickness?
4. Can the author compare the electrochemical analysis with pristine CNTs (CNTs before annealing)? I think this could be a good comparison to show the improvements pre-annealing vs post-annealing.
Author Response
- Added pictures and yields of MWNTs preparation using CVD method, see file for details.
- The mass of each substance has been added, see the document for details.
- The thickness of the coating is 0.019mm, see the document for details.
- Electrochemical data of the cell (NCM-CNT) prepared by adding impurity-free pristine MWNTs (slurry) as a conductive additive and adding its corresponding analysis are detailed in the manuscript (yellow section).

Reviewer 2 Report
The reviewed manuscript entitled: "High-Temperature Annealed Multi-Walled Carbon Nanotubes as High Performance Conductive Agents for LiNi0.5Co0.2Mn0.3O2 Lithium-Ion Batteries" deals with exploring high temperature annealed and demetallized MWCNTs and their utilization in preparation of conductive slurry with the use of micro sand milling technique. The obtained CNT slurry is further used in as conductive agents in LiNi0.5Co0.2Mn0.3O2 (NCM) cathode materials by sand-mill dispersion and its performance of lithium-ion batteries was investigated. The results obtained showed that the high temperature annealing can effectively remove the residual metal impurities from MWNTs and the defects in MWNTs gradually decreases as the temperature rises. The MWCNTs used as conductive agents in LiNi0.5Co0.2Mn0.3O2 cathode materials show excellent battery behaviors, which really provide a new insight for the development of high-performance novel conductive agent in lithium-ion batteries.
Few critical remarks:
In the Experimental part, please unbold the labelled abbreviations of CNT-2000 etc.
I suggest combining Fig. 8 and 9 in one Figure and pls do not end the manuscript with blanc Figure and the following Conclusions. In the Conclusion part end there is wrongly put irrelevant text.
Author Response
1. Already modified, please see the document for details.
2. Figures 8 and 9 have been merged and the irrelevant text at the end of the conclusion has been removed, plwase see document for details.
